# The Ectodomains of rBAT and 4F2hc Are Fake or Orphan α-Glucosidases

**DOI:** 10.3390/molecules26206231

**Published:** 2021-10-15

**Authors:** Joana Fort, Adrià Nicolàs-Aragó, Manuel Palacín

**Affiliations:** 1Laboratory of Amino Acid Transporters and Disease, Institute for Research in Biomedicine (IRB Barcelona), The Barcelona Institute of Science and Technology (BIST), Baldiri Reixac 10, 08028 Barcelona, Spain; adria.nicolas@irbbarcelona.org (A.N.-A.); manuel.palacin@irbbarcelona.org (M.P.); 2CIBERER (Centro Español en Red de Biomedicina de Enfermedades Raras), 08028 Barcelona, Spain; 3Department of Biochemistry and Molecular Biomedicine, Universitat de Barcelona, 08028 Barcelona, Spain

**Keywords:** SLC3, alpha-amylase, alpha-glucosidase, structure, transporter, ancillary protein, scaffold protein, ectodomain, *N*-glycosylation

## Abstract

It is known that 4F2hc and rBAT are the heavy subunits of the heteromeric amino acid transporters (HATs). These heavy subunits are *N*-glycosylated proteins, with an N-terminal domain, one transmembrane domain and a bulky extracellular domain (ectodomain) that belongs to the α-amylase family. The heavy subunits are covalently linked to a light subunit from the SLC7 family, which is responsible for the amino acid transport activity, forming a heterodimer. The functions of 4F2hc and rBAT are related mainly to the stability and trafficking of the HATs in the plasma membrane of vertebrates, where they exert the transport activity. Moreover, 4F2hc is a modulator of integrin signaling, has a role in cell fusion and it is overexpressed in some types of cancers. On the other hand, some mutations in rBAT are found to cause the malfunctioning of the b^0,+^ transport system, leading to cystinuria. The ectodomains of 4F2hc and rBAT share both sequence and structure homology with α-amylase family members. Very recently, cryo-EM has revealed the structure of several HATs, including the ectodomains of rBAT and 4F2hc. Here, we analyze available data on the ectodomains of rBAT and 4Fhc and their relationship with the α-amylase family. The physiological relevance of this relationship remains largely unknown.

## 1. Introduction

rBAT (SLC3A1, also named D2 and NBAT) and 4F2hc (SLC3A2, also named CD98hc and FRP, for fusion regulatory protein) are the heavy subunits of the heteromeric amino acid transporters (HATs) [1] and they are linked to an _L_-amino acid transporter (LAT) subunit from the SLC7 gene family by a disulfide bridge (Figure 1a). These heavy subunits are type II membrane proteins with a cytoplasmatic N-terminus, one transmembrane domain (TM) and a bulky extracellular *N*-glycosylated C-terminal domain (50–60 kDa) or ectodomain (ED) (Figure 1a). In humans, six LAT transporters (LAT1, LAT2, y^+^LAT1, y^+^LAT2, asc-1, and xCT) heterodimerize with 4F2hc and two (b^0,+^AT and AGT1) with rBAT, thereby rendering a range of different amino acid transport activity [2]. The heavy subunits of HATs appeared in metazoans with a primitive heavy subunit and this evolved into the vertebrate proteins 4F2hc and rBAT [3,4]. These two human proteins share approximately 25% amino acid sequence identity. Surprisingly, their bulky extracellular domains share similar sequence identity or even greater, around 25% and 40% for 4F2hc-ED and rBAT, respectively, with members of α-amylase family, specifically from the GH13 family [5,6].

The heavy subunit in HATs is required for the correct localization of the functional holotransporter to the plasma membrane, whereas the light subunit is responsible for the transport function (reviewed in [10,11]). Nevertheless, the heavy subunit can modulate the transport function. In this regard, 4F2hc-LAT1, 4F2hc-LAT2, LAT1 and LAT2 expressed in *Pichia pastoris* demonstrate that the presence of 4F2hc modulates substrate affinity and specificity in the light subunit-associated transport [12]. Additionally, one report showed differences in the requirement of the ectodomain of 4F2hc (4F2hc-ED) for the plasma membrane localization of some light subunits. Working with domain truncations of 4F2hc revealed that the presence of the ED is essential for LAT2 and y+LAT2 but not for LAT1 [13]. Finally, studies with detergent-solubilized LAT2 revealed increased stability when the transporter was incubated with the recombinant 4F2hc-ED [14].

Furthermore, 4F2hc, with its associated light chains, is an almost ubiquitously expressed protein playing a central role in cellular nutrition, redox homeostasis and nucleotide availability, being a modulator of cell proliferation [15,16,17]. Indeed, 4F2hc together with light subunits LAT1 and xCT is overexpressed in different types of cancers [18,19,20] and in activated lymphocytic cells [21], thereby supporting that this heavy subunit is involved in cell growth. Moreover, there is a functional interaction between 4F2hc and β1A and β3 integrins, which modulates integrin-related signaling and is essential for integrin-dependent cell spreading, migration and tumor progression [22,23,24,25,26]. Mapping of the interactions of 4F2hc with integrins using protein chimeras with the unrelated glycoprotein CD69 demonstrated that the key protein parts are the TM and the cytosolic N-terminal domain [22,24]. Nevertheless, the 4F2hc-ED might also have a role modulating the β1 integrin function and tumorigenicity [27]. Additionally, the interaction of lectin galectin-3 with the extracellular domain of 4F2hc has been described to play a role in trophoblast formation by cell fusion in the placenta [28].

The heavy subunit rBAT has not been related to functions other than the targeting of b^0,+^AT to the plasma membrane, where the holotransporter transports neutral (0) and basic (+) amino acids. rBAT/b^0,+^AT expression is restricted to the apical domain of the plasma membrane of the epithelial cells in the small intestine and in the renal proximal tubule, whereas 4F2hc, together with corresponding LAT transporters, is located in the basolateral plasma membrane in epithelial cells [29], thereby pointing to the participation of the heavy subunit in the location of the holotransporter in these polarized epithelial cells. Importantly, some mutations in rBAT cause type A cystinuria, an inherited aminoaciduria caused by the loss of function of the rBAT/b^0,+^AT transporter in the epithelial cells [30] of the kidney and intestine. The studied mutations in rBAT-ED cause a defect in the biogenesis, stability and arrival to the plasma membrane of the holotransporter [31].

Another feature associated with heavy subunits is that, in native tissues, 4F2hc-associated transporters are single heterodimers, whereas rBAT/b^0,+^AT is a dimer of heterodimers ([rBAT/b^0,+^AT]_2_), despite the heterodimer being the functional transport unit in the membrane [32]. Experiments with chimeras joining rBAT to 4F2hc-associated light subunits show the formation of a functional dimer of heterodimers, indicating that the heavy subunit governs the oligomerization state [32]. In this regard, cryo-EM structures have demonstrated that the interactions of dimers of heterodimers in rBAT-b^0,+^AT occur through rBAT-ED (Figure 2a,b) [9,33,34]. The cytosolic *N*-terminus of rBAT has not been solved, leaving open the possibility of further interactions in this protein part. Although homodimers of 4F2hc can be found upon overexpression in cultured cells and in one of the crystal structures of recombinant 4F2hc-ED [5], a dimer of heterodimers never has been described in vivo or in any further 4F2hc structures (Table 1).

The ectodomains of 4F2hc and rBAT show sequence homology with α-amylase enzymes, specifically with the oligo-1,6-glucosidase subfamily in glycosyl hydrolases family 13 (GH13) [4,40,41]. The large α-amylase family comprises more than 30 distinct substrate specificities, but its members share defining common features: (i) they catalyze the action on α-glucosidic bonds through a retaining reaction mechanism, (ii) the catalytic domain is a (β/α)8-barrel fold (i.e., TIM-barrel), (iii) they have between four and seven conserved regions mainly in the β strands of the barrel and (iv) the conserved catalytic site is formed by an aspartate (nucleophile), a glutamate (proton donor) and another aspartate (transition-state stabilizer) placed in the vicinities of β4, β5 and β7 strands [40,42,43,44,45,46,47,48,49]. The GH13 family includes members from several enzyme activities as α-amylases, α-glucosidases, α-1,4-glucan branching enzymes, pullulanases, cyclodextrin glucanotransferases, 4-α-glucanotransferases, oligo-α-1,6-glucosidases. Due to sequence and structural homology the ectodomains 4F2hc and rBAT have been classified in the Carbohydrate-Active Enzymes database into the GH13 family [41] in the oligo-1,6-glucosidase subfamily [50] and, more specifically, 4F2hc in GH13_34 and rBAT in GH13_35 [51].

In this review, we update the structural and functional information about the 4F2hc-ED and rBAT-ED, their relationship with α-glucosidases and we also discuss their possible physiological roles.

## 2. Structural Information about 4F2hc and rBAT

The cryo-EM structures of 4F2hc and rBAT (Table 1) within the respective holotransporters revealed some clues about the relationship between the two subunits. Regarding the heterodimer interface, the heavy subunit interacts extensively with its respective light subunit via several regions. A disulfide bond is present in the neck, between the TM and the ED, in the heavy subunit and the extracellular loop two (EL2), between TM3 and TM4, in the light subunit (Figure 1a and Figure 2a). In addition, there is a tight hydrophobic interaction in the membrane region between the TM of the heavy subunit and the TM4 in the light subunit (Figure 1a). In the cytoplasm, there is an interaction between the α-helix located in the N-terminal of the heavy subunit and the C-terminal of the light subunit (Figure 1a). Due to the lack of structural information for the first residues of the *N*-terminus of the heavy subunits other interactions cannot be disregarded.

The ED of the heavy subunits is located above the light subunit in all the solved structures (Table 1 and Figure 1a and Figure 2a), showing many polar interactions in addition to the disulfide bond between the two subunits. However, some interfaces between the N-terminal (β/α)_8_-barrel loops and extracellular loops of the light subunits are variable. In the 4F2hc-LAT2 structure, with LAT2 in an inward-facing conformation (i.e., the substrate vestibule open to the cytosol), these interactions are not present. On the other hand, the interfaces between 4F2hc-ED and LAT1 in an inward-facing conformation [37] are only partially conserved when LAT1 adopts a different outward-facing conformation [8]. Overall, the position of the 4F2-ED with respect to the different light subunits is very similar in 4F2hc-LAT1 and 4F2-LAT2, where the light subunits show an inward-open conformation (reviewed in [52]), but rotates approximately 5.3° in structures of 4F2-LAT1 in outward-facing conformations [8]. Moreover, in the 4F2hc-xCT structure, with xCT showing an inward-facing conformation, there is a displacement of ~20° vertically away from the entrance of the amino acid binding site in the light subunit [38]. The relative position of the bulky ED above the light subunit has been proposed using a low-resolution structure of the heterodimer 4F2hc-LAT2, with docking and crosslinking experiments between both subunits [14]. The crosslinking results do not fit with the 4F2hc-LAT2 cryo-EM structure [7] or any structure of 4F2hc with xCT or 4f2hc-LAT1. However, 4F2hc in membranes of mammalian cells might explore different conformations than in cryo-EM studies, where the transporter is solubilized in detergent micelles. More structures of different 4F2hc holotransporters and in a lipid environment are needed to define the conformational landscape of 4F2hc-ED interactions with the light subunits in HATs.

The 4F2hc-ED is linked to the TM through a neck of 11 residues (Figure 1a,b). This neck interacts with the globular ED in a very similar way in all the solved structures. This neck is shorter in rBAT-ED, having only six residues (Figure 1c and Figure 2a). This implies differences between the heterodimerization surface of the rBAT-ED and 4F2hc-ED with their respective light subunits. Cryo-EM structures showed a displacement of ~40 Å between them, creating different interfaces [34]. Moreover, the bottom surface of the rBAT-ED is mostly negatively charged, whereas that of 4F2hc-ED is positively charged, thus pointing to distinct electrostatic interactions [5,34].

In summary, the relative position of the ED of heavy subunits with respect to the light subunit in HATs is seen to be slightly variable in the structures, depending on the light subunit and also depending on the conformational state of this light subunit. In this regard, the differences in interaction interfaces suggest a possible stabilizer role of the ED but allowing a certain degree of flexibility between the two subunits of the heterodimer, which may be needed for protein function.

## 3. Structure of the Ectodomains of 4F2hc and rBAT

The first available structure for the HAT family was the ectodomain of 4F2hc (Table 1) [5]. As predicted by sequence homology, the structure of 4F2hc-ED shows the topology and multidomain organization of α-amylases (Figure 1b). To date, seventeen 4F2hc structures have been deposited in the RCSB database (Table 1). Five 4F2hc-ED structures have been solved by X-ray crystallography and one by cryo-EM, and eleven structures of full 4F2hc heterodimerizing with different LAT transporters and ligands have been achieved by cryo-EM (Table 1). Superimposition of all the available atomic structures of the EDs of 4F2hc (Table 1) reveals high similarity for this extracellular domain, with a calculated Rooth Mean Square Distance (RMSD) below 0.7 A^2^, and without significant changes in any region of the structure.

Furthermore, 4F2hc-ED contains the so-called domain A in α-amylases, with (β/α)_8_-barrel fold, linked to the TM domain by the eleven-residue neck indicated above (Figure 1a,b). At the end of domain A, a six-residue helix connects to the domain C (Figure 1b) that is consistently present in α-amylases, although presenting a variable β-sandwich topology [40].

The first structures of human rBAT forming a dimer of heterodimers with b^0,+^AT and a single homodimer of rBAT (Table 1, Figure 2) have also recently been deposited in the Protein Data Bank. The ectodomain of rBAT has a similar domain composition to 4F2hc with domain A and C, but, additionally, it contains domain B, which is also common in the loop between Aβ3 and Aα3 in the (β/α)_8_-barrel of the α-amylase family (Figure 1d). This loop has three β-strands and one α-helix, and it contacts domain A through a cluster of hydrophobic residues. Finally, rBAT contain a C-terminus cyclic C-terminal β-hairpin stabilized by a disulfide bridge (Cys^673^-Cys^685^) protruding from domain C (Figure 1c,d). This C-terminal small hairpin is key for the maturation of human rBAT protein as tested in mammal cells [53].

The domain B of rBAT is similar to that of oligo-1,6-glucosidases as predicted by sequence homology [40]. The loop Aβ3-Aα3 is stabilized by a disulfide bridge between Cys^242^ and Cys^273^, which had already been predicted by modeling based on oligo-1,6-glucosidase from *B. cereus* [5] and biochemically demonstrated [54]. Interestingly, a Ca^2+^ binding site is present in this loop in the cryo-EM structures (Figure 1c,d) [9,33,34]. Indeed, binding to Ca^2+^ is important for the function and stability of many members of the α-amylase family [55,56]. Moreover, *Anoxybacillus* sp. SK3-4 amylase (ASKA) has a conserved Ca^2+^ site in the same position as rBAT, stabilizing domain B [57]. In detail, the Ca^2+^ binding site in rBAT-ED stabilizes the interaction of the loop Aβ3-Aα3 with the next one in the (β/α)_8_-barrel, the Aβ4-Aα4 (Figure 1d). Due to these strong interactions, domain B could be considered to be formed by both loops as subdomains B-I and B-II, since Ca^2+^ interacts with and stabilizes them as a single domain above the (β/α)_8_-barrel (Figure 1c,d) [34]. Subdomain B-II is partially responsible for the homodimerization between two molecules of rBAT, as shown in the structures of the dimers of heterodimers or superdimers and in the homodimeric rBAT structure (Table 1, Figure 2a and red spheres in b) [9,33,34]. This superdimerization is crucial for heterodimer maturation, as demonstrated in experiments using mutations in residues in the homodimerization interphase (Figure 2b) [34]. In this regard, mutations that affect the stability of domain B, such as cystinuria-causing mutation T216M or the residues responsible of Ca^2+^ binding, confirm that this domain B is crucial for superdimerization and therefore heterodimer maturation [31,34]. There is another homodimerization interface localized in Aα4 and the loop between Aβ5 and Aα5 in domain A (blue spheres in Figure 2b), but no other interactions are present in the structure between both rBAT molecules outside of the ectodomain.

Most of the mutations in rBAT described to cause cystinuria are located in the ED [2]. Biogenesis of human rBAT-b^0,+^AT with mutated versions on rBAT protein T216M, R365W, M467T or M467K showed impaired biogenesis being expressed in mammal cells [31]. In this regard, recombinant rBAT-b^0,+^AT with mutations V183A, T216M, M467T and L678P in rBAT-ED showed decreased stability and the recombinant protein could not be purified properly [9]. With respect to the structure, T216 is located in the middle of the hydrophobic interaction between domains A and B and M467 is at the interface of domains A and C, explaining why changes in these residues might destabilize the protein. On the other hand, R365 is stabilizing loop Aα4-β5, which interacts with the light subunit, thus explaining why the R365W mutation might affect the transport characteristics of the b^0,+^AT system [58]. Finally, L678 is in the C-terminal β-hairpin (Figure 1c), which interacts with the light subunit and the lipid bilayer in structures [9,33,34] and is also shown to be important for the biogenesis of the rBAT-b^0,+^AT heterodimer [53].

The cryo-EM structures of rBAT and 4F2hc expressed in mammal cells showed the characteristic glycosylations in some Asn residues of the ED. The 4F2hc-ED shows four *N*-glycosylated positions in domain A (Asn^264^ in Aα4, Asn^280^ just before Aα5, Asn^323^ in Aα6, and Asn^405^ in loop Aβ8-Aα8) (Figure 1b). In contrast, human rBAT has five *N*-glycans (Asn^261^ in domain B, Asn^332^ in loop Aβ4-Aα4, Asn^495^ and Asn^513^ in the loop Aβ8-Aα8 in the domain A and Asn^575^ in the domain C) (Figure 1c). These positions are not conserved between both proteins (Figure 1b,c). Indeed, orthologous rBAT proteins have different *N*-glycosylation patterns, as reflected by the structure of ovine rBAT, which has six *N*-glycosylation sites, of which only three are shared with the human protein [34]. Unexpectedly, rBAT *N*-glycosylation in Asn^575^ is the only one that is needed for the biogenesis of the heterodimer [53]. However, the same glycosylation in the Asn^575^ residue is not conserved in the ovine (non-glycosylated) as in human (glycosylated) rBAT-ED [34]. *N*-glycosylation is also observed in some α-amylases of mammals [59], plants [60], and even fungi [61]. *N*-glycosylation in these enzymes is related to stability and also serves as a modulators of the enzymatic activity, but no structural pattern has been found between them [62].

The structures of 4F2hc-ED produced in *E. coli* (without *N*-glycosylations) or in mammalian cells (*N*-glycosilated), do not show large differences, even near the glycosylation sites, thus indicating that *N*-glycosilations in 4F2hc are not needed for proper protein folding. Moreover, 4F2hc-ED produced in *E. coli* shows high stability [63] and even confers stability to LAT2 when it is solubilized in detergent [14].

## 4. Sequence and Structure Relationship of rBAT-ED and 4F2-ED with α-Glucosidases

The rBAT-ED (residues 117–685) shares around 40% sequence identity (SI) with maltase-1 isoform-X3 from *Cryptotermes secundus* and 4F2hc-ED (residues 219–631) shares 26% SI with α-amylase of *Pseudomonas indica* (α-amylases with higher scores in BLAST-P against nr-database). The structure conservation between these ectodomains and α-amylases is even higher, as can be seen superimposing the structures of rBAT-ED and of the α-glucosidase Cqm1 (PDB ID 6K5P) [64],which that show only a difference of 1.06 Å^2^ RMSD, sharing 40% SI. Indeed, the crystal structure of 4F2-ED was solved by molecular replacement using as a model an oligo-1,6-glucosidase structure (PDB-ID 1UOK), sharing above only 25% sequence identity [5]. Detailed sequence evolutionary studies supports a greater bigger distance of 4F2hc-ED from an ancient glucosidase-like protein closer to rBAT and members of the GH13 family [4].

Importantly, most of the defining features of the α-amylase family are present in 4F2hc-ED and especially in rBAT-ED. The EDs of both proteins have a (β/α)_8_-barrel fold in domain A followed by a domain C with a β-sandwich fold (Figure 1b,c). rBAT-ED also has a domain B similar to that of oligo-1,6-glucosidases [40] (Figure 1c,d). In addition, rBAT-ED conserves the common catalytic site formed by a β4-strand aspartate (nucleophile), a β5-strand glutamate (proton donor) and a β7-strand aspartate (transition-state stabilizer) in positions Asp^314^, Glu^384^ and Asp^449^ (Figure 3d), as can be seen in comparison with the structures of the catalytic site of Halomonas sp. H11 α-glucosidase (HaG) complexed with maltose and oligo-1,6-glucosidase from *B. cereus* (Figure 3d). Moreover, other conserved important residues for fixing the glucose residue at subsite −1, such as His^105^,Asp^62^ and Asp^333^ in HaG, are conserved, whereas Arg^200^ or His^332^ (in HaG) are not conserved in human rBAT-ED (Figure 3d). Phe^297^ in HaG, which is also not conserved in rBAT-ED, has been related to the recognition of the nonreducing-end glucosyl residue [65]. Finally, the location of Ala^315^ in rBAT-ED (not shown in Figure 3d) is hypothesized to be compatible with the specific hydrolysis of α-1,4-glycosidic bonds [66].

Interestingly, the loop Aβ2-Aα2 in rBAT, where is located Asp^62^ in HaG, has a conformation not compatible with the position of maltose in the binding site of HaG (Figure 3d). The conformation of this loop is similar in structures of HaG with or without substrates (PDB: 3WY4 and 3WY1, respectively) and even in oligo-1,6-glucosidase of *B. cereus* (Figure 3d), which has different substrate specificity. In contrast, all the available structures of rBAT (Table 1) present the displaced conformation in the Aβ2-Aα2 shown in Figure 3d, suggesting a structural difference in the site of rBAT with respect to α-glucosidases and oligo-1,6-glicosidases. Furthermore, the catalytic triad is not fully conserved in all the putative rBAT orthologues [4]. Indeed, other conserved sequence regions (CSR) are more conserved than these catalytic residues among the vertebrate rBAT homologues [4]. The fact that these key residues are not preserved by negative selection suggests that the α-amylase catalytic activity is not important for rBAT function, at least in some vertebrates species [6]. All these differences might reflect changes in substrate recognition and specificity in rBAT-ED that can even lead to a total lack of α-glucosidase or oligo-1,6-glucosidase activity. Computational studies of molecular dynamics, docking and binding with different glycosidic substrates would be pertinent to predict whether this special conformation in the loop Aβ2-Aα2 can allow some flexibility or other substrates to bind.

Finally, the catalytic cleft in 4Fhc-ED and rBAT-ED is similar to that present in α-glucosidase HaG (Figure 3a–c). As expected, in 4F2hc, the accessibility of the cleft is higher because of the lack of domain B above domain A. The presence of a similar cavity suggests that some α-glucosidase-related function could be retained by these ectodomains, with or without catalytic activity.

## 5. Efforts to Prove Any rBAT-ED and 4F2-ED Glucosidase-Related Activity

Despite the strong evidence of the sequence and structural homology of the EDs of rBAT and 4F2hc with α-amylases, no α-glucosidase activity has been demonstrated for either protein [5,33]. In 4F2hc-ED, the three catalytic amino acids (Asp^206^, Glu^230^ and Asp^297^ in Taka amylase A), common in all the α-amylase family, are missing. Therefore, 4F2hc-ED is not expected to have α-glucosidase activity. In this regard, null detection of activity was found in a detailed screening using d-glucose, d-galactose or d-mannose derivatives of 4-methylumbelliferone as substrates [5]. In all, these results support the notion that 4F2hc-ED does not have catalytic α-glucosidase activity but might still retain some glycosidase-like binding activity, as is shown for some other α-amylase family members, such as Chitinase-like protein 3 from GH18 [69] and Edem1 (ER degradation-enhancing α-mannosidase-like protein 1) from the GH47 family [70]. Despite this, studies of carbohydrate binding with commercial glycochips from Glycominds (48 different glycans) and from the Glycomics Consortium (260 different glycans) failed to detect carbohydrate binding by human 4F2hc-ED [5]. It has also been proposed that some proteins or glycoproteins bind to the wide cleft of 4F2hc-ED, especially Ig-like containing proteins because of the homology with tendamistat, an α-amylase inhibitor that interacts with the catalytic cleft [71]. Moreover, several proteins with Ig-like domains, such as galectin-3 [72], ICAM-1 [73], CD-147 [74] and CEA-CAM-1 [75], have been proposed to interact with 4F2hc-ED. However, no structural information for these complexes has been published to date.

In contrast to 4F2hc-ED, rBAT-ED conserves the three catalytic residues of the α-amylase family (Figure 3d). Nevertheless, heterodimer rBAT-b^0,+^AT solubilized in detergent (glyco diosgenin 0.02%) failed to demonstrate α-glucosidase activity [9]. Studies with wider range of substrate species should be conducted in order to disregard any other GH13-related activity. To the best of our knowledge, no other functions for rBAT-ED, such as binding to glycans, or any other in relation to the α-amylase structure, have been proposed or tested in rBAT so far.

## 6. Conclusions

In conclusion, 4F2hc and rBAT, the heavy subunits of heteromeric amino acid transporters, have an ED with sequence and structure homology with α-glucosidases that can be classified as members of the GH13 family.

rBAT-ED and 4F2hc-ED have a (β/α)_8_ barrel as a domain A followed by domain C with a β-sandwich fold. In addition, rBAT has also domain B protruding from domain A and formed by two different loops, Aβ3-Aα3 and Aβ4-Aα4. This domain B is stabilized by a disulfide bridge and a Ca^2+^ ion, similarly to in α-amylases. This domain B is key for the functional oligomeric state of rBAT-b^0,+^AT as superdimers.

The characteristic active site of α-amylases, including key catalytic residues, is fully conserved in rBAT-ED and not in 4F2hc-ED. However, α-glucosidase-related activity (catalytic or lectin-like) has not yet been proven for any of these two heavy subunits. Further experiments are needed to unravel the physiological role of the homology of rBAT-ED and 4F2hc-ED with α-amylase family members.

## Figures and Tables

**Figure 1 molecules-26-06231-f001:**
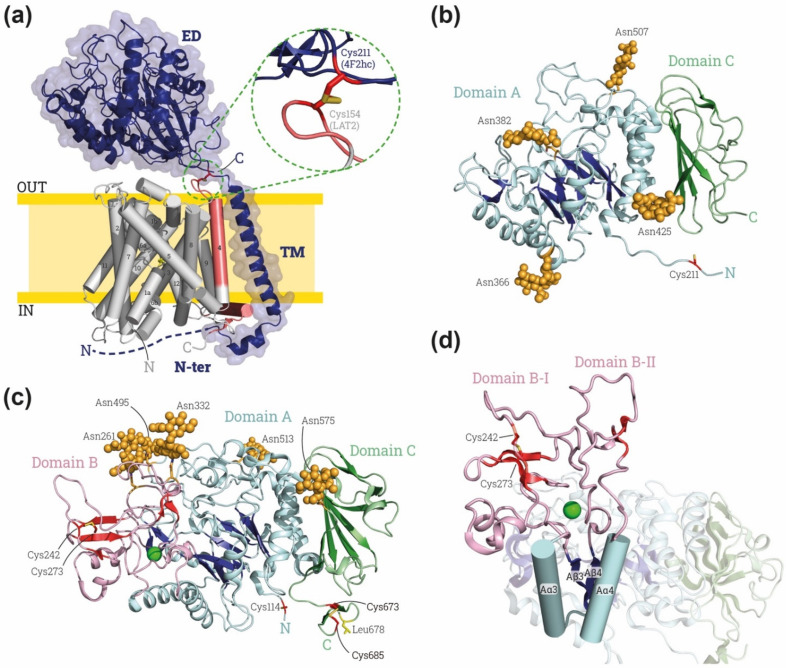
Overview of HAT structures. (**a**) Representative HAT structure. Cryo-EM structure of human 4F2hc-LAT2 (PDB: 7CMI [7]) is shown: heavy subunit 4F2hc is indicated in dark blue and by blue cartoon with a transparent surface, and light subunit LAT2 is indicated in gray and red cylinders. The transmembrane (TM) domains of LAT2 are numbered from the N to the C-terminus. On the right, close-up for the disulfide bond between heavy and light subunit. Heavy subunit contains an ectodomain (ED), a transmembrane domain (TM) and an N-terminus domain (N-ter). Domains of LAT-2 interacting with heavy subunit are colored in red. (**b**) Human 4F2hc-ED structure, PDB: 7DSK [8]. The four glycosylation moieties are shown in light orange spheres, and subdomains A and C in blues and greens, respectively, with brighter colors indicating the β-sheets. The cysteine involved in the disulfide bridge with LATs (Cys 211 in human 4F2hc) is shown in red at the *N*-ter. (**c**) Human rBAT-ED structure, PDB: 6LI9 [9]. The five glycosylation moieties are labelled by light orange spheres, and subdomains A, B and C in blue, red and green, respectively, with brighter colors indicating the β-sheets. The cysteine involved in the disulfide bridge with LATs (Cys 114 in human rBAT) is shown in red at the N-terminus of rBAT-ED. L678 is colored in yellow and Ca^2+^ atom is represented by green ball. (**d**) Detailed view of the domain B with subdomains BI and BII (loops Aβ3-Aα3 and Aβ4 and Aα4, respectively) stabilized by a Ca^2+^ (green ball). Images were created with the PyMol Molecular Graphics System, Version 2.1.4., Schrödinger, LLC.

**Figure 2 molecules-26-06231-f002:**
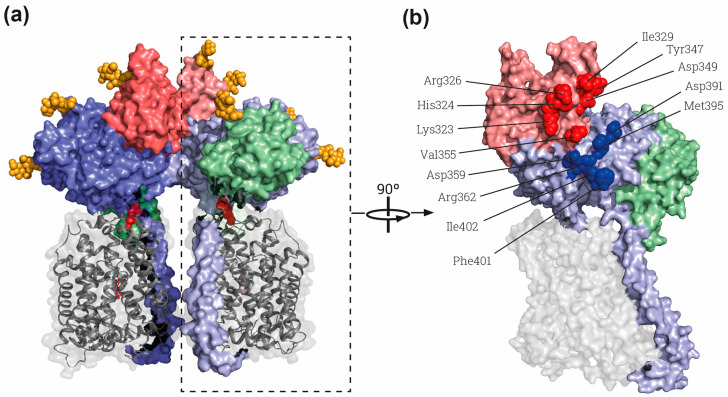
The rBAT-ED is responsible for dimerization. (**a**) General view of the whole superdimer of rBAT-b^0,+^AT (PDB: 6LI9) [9]. Dashed rectangle indicates the heterodimer represented in B). (**b**) Lateral view of the interaction side in the heterodimer. Residues involved in the rBAT-rBAT homodimerization within the domains B and A of the ED are labelled by red and blue spheres, respectively. Each heterodimer dimer is shown in different intensity of same color, gray for b^0,+^AT and blue, red and green for rBAT domains TM and A, B and C, respectively. The cysteines involved in the disulfide bridge between subunits and glycosylation moieties are represented by red and orange spheres respectively. Images were created with the PyMol Molecular Graphics System, Version 2.1.4., Schrödinger, LLC, New York, USA.

**Figure 3 molecules-26-06231-f003:**
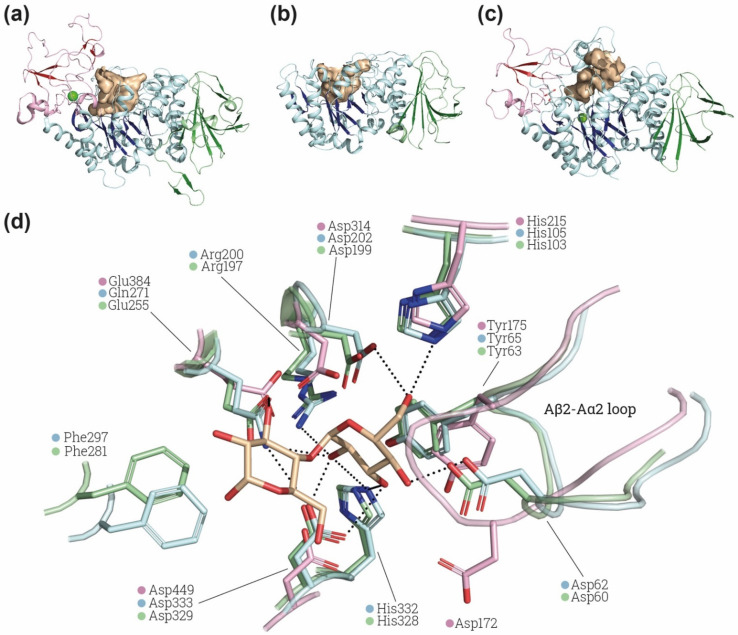
α-glucosidase active site comparison. Structural comparison of the substrate cavity between rBAT-ED (PDB: 6LI9 [9]) (**a**), 4F2hc-ED (PDB: 6IRT [37]) (**b**) and *Halomonas* sp. H11 α-glucosidase (HaG) (PDB: 3WY4 [65]) (**c**). The cavities of all the structures are shown in light brown and domains A, B and C in blue, red and green, respectively, β-sheets are more brightly colored. Ions (Ca^2+^ and Mg^2+^ for rBAT and HaG respectively) are indicated in light green. Cavity area and volume were also measured using the Computed Atlas of Surface Topography of proteins (CASTp) plugin [67] on PyMol. Calculated areas for cavities are 633.32 Å^2^, 802.78 Å^2^ and 802.78 Å^2^ for 4F2hc, rBAT and HaG, respectively. (**d**) Detailed view of HaG (light blue) (PDB: 3WY4) and oligo-1,6-glucosidase of *B. cereus* (light green) (PDB: 1UOK [68]) binding pocket and the conserved residues found in rBAT (light pink) (PDB: 6LI9). Maltose is labelled in light brown. Non-conserved residues of rBAT are not shown. Images were created with the PyMol Molecular Graphics System, Version 2.1.4., Schrödinger, LLC.

**Table 1 molecules-26-06231-t001:** HAT heavy subunit structures deposited in the Protein Data Bank (RCSB). Structures are grouped by the protein solved. The Uniprot code, solved region (aa), technique of resolution (Tech.), overall resolution (Å), Protein Data Bank entry codes (PDB), details of the structure (such as ligands, antibodies (Fab), nanodisc (ND) reconstitution) and references (Ref.) are also indicated.

Name of Structure	Uniprot	Residues	Tech.	(Å)	PDB	Structure Details	Ref.
human 4F2hc-ED	P08195	212–630	X-ray	2.10	2DH2	Monomer	[5]
	P08195	212–630	X-ray	2.80	2DH3	Homodimer	[5]
	P08195	212–630	X-ray	1.80	6S8V	Monomer-Antivalin P3D11	[35]
	P08195	210–630	Cryo-EM	4.10	6JMR	+ Fab HBJ127 + Fab MEM-108	[36]
mouse 4F2hc-ED	P10852	105–526	X-ray	2.75	6SUA	Monomer-Lipocalin C1B12	[35]
	P10852	108–526	X-ray	2.10	6I9Q	Monomer	[35]
human 4F2hc- LAT1	J3KPF3	163–631	Cryo-EM	3.50	6IRT	(A36E) + BCH*	[37]
	J3KPF3	163–631	Cryo-EM	3.30	6IRS	(A36E)	[37]
	P08195	180–630	Cryo-EM	3.31	6JMQ	+ MEM-108 Fab	[36]
	J3KPF3	161–631	Cryo-EM	2.90	7DSK	+ JX-075*	[8]
	J3KPF3	161–631	Cryo-EM	2.90	7DSL	+ JX-078*	[8]
	J3KPF3	161–631	Cryo-EM	3.10	7DSN	+ JX-119*	[8]
	J3KPF3	162–631	Cryo-EM	3.40	7DSQ	+ Diiodo-Tyr*	[8]
human 4F2hc-xCT			Cryo-EM	3.40	7P9V		[38]
	P08195	162–630	Cryo-EM	6.20	7CCS	(consensus mutated)	[39]
human 4F2hc-LAT2	J3KPF3	163–631	Cryo-EM	2.90	7CMI	+ l-Leu	[7]
	J3KPF3	163–631	Cryo-EM	3.40	7CMH	+ l-Trp	[7]
ovine rBAT-ED	W5P8K2		Cryo-EM	2.68	7NF7	Monomer	[34]
human rBAT homodimer	Q07837	92–685	Cryo-EM	2.80	6YUZ	Homodimer	[33]
human rBAT-b^0,+^AT heterotetramer	Q07837	63–685	Cryo-EM	2.70	6LID		[9]
	Q07837	63–685	Cryo-EM	2.30	6LI9	+ l-Arg	[9]
	Q07837	92–685	Cryo-EM	2.90	6YUP		[33]
ovine rBAT-b^0,+^AT heterotetramer	W5P8K2		Cryo-EM	2.86	7NF8	Reconstituted in ND	[34]
ovine rBAT-b^0,+^AT heterodimer	W5P8K2		Cryo-EM	3.05	7NF6	Reconstituted in ND	[34]

## Data Availability

Not applicable.

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
