# Peer review of "The Ectodomains of rBAT and 4F2hc Are Fake or Orphan α-Glucosidases"

_molecules, 2021, doi:10.3390/molecules26206231_

Round 1

Reviewer 1 Report

Although the amino acid residues resembling the catalytic and substrate binding residues of enzymes from the family GH13 are present in the human rBAT protein, at corresponding positions, there are significant differences in the spatial orientation of these residues between human rBAT and GH13 proteins. Addition of a second GH13 enzyme into the structural comparison of the substrate cavity (Fig. 3) could visually show how conserved and important orientation of these residues for proper enzymatic function is and how much are key residues of the catalytic pocket shifted in the tertiary structure(s) of the human rBAT protein. Moreover, as the comparison (Gabrisko & Janecek. FEBS J. 2009. 276(24):7265-7278. doi: 10.1111/j.1742-4658.2009.07434.x) of the primary structures of the putative rBAT homologues from selected vertebrate species shows, these key catalytic and substrate binding residues are not conserved in all studied species. Actually, no important residue is conserved in all studied species. Also, the selection pressure analysis shows that the key residues are not preserved by negative (purifying) selection. Observed lack of conservation could imply, that these key residues are not important for function of rBAT in some or maybe all vertebrate species. Interestingly, the conserved sequence regions (CSR) I (strand β3) and VII (strand β8) are more conserved among the vertebrate rBAT homologues, than the residues of the catalytic pocket. Important part of the origin and function puzzle of the HAT proteins are also HAT proteins of invertebrates. Their specificity and relationship to the vertebrate HATs remains mostly unknown.

Author Response

Dear reviewer,

It has been very useful your review of our manuscript. We accepted all your suggestions that improved Figure 3 and the comparison with other rBAT orthologues in the text. It has been an excellent contribution.

Changes applied:

We added the binding pocket of oligo-1,6-glucosidase of B-cereus in Figure 3D following the advice of the reviewer.

We added three sentences about the comparison of the conservation of the catalytic residues in other rBAT orthologues (highlighted in green, lines 305-310).

Thank you,

Joana Fort

Reviewer 2 Report

Fort et al describe numerous subtle structural details of the amino acid transporters related to glycoside hydrolase family 13. This is an excellent work and very useful for readers. Some essentially very small often cosmetics points to be taken into account by the authors are given below.  

  1. Line 83, elsewhere in the manuscript “0.+” is superscript to b? Please also for the readers define what is b0,+
  2. The term superfamily for GH13 (line 105) is not really appropriate. Secondly, “subgroup” should be called “subfamily”
  3. In line 109 it would be good to rephrase a bit “..catalyze the action on α-glucosidic..”
  4. Lines 112-113, please note that some of the three conserved catalytic residues may not be in the β-strand proper, but at its C-terminal end starting the loop to the next a-helix of the TIM barrel
  5. Lines 129-133 are not so easy to follow, may these be rephrased. How many light subunits are there in the structure of the transporter?
  6. ASKA is not understood by readers, please explain and provide a recognized name.

Minor:

  1. In the abstract please use small cap L to mark the specificity for stereoisomers
  2. In the abstract, correct to 50-60 kDa
  3. Line 66, correct to “revealed”
  4. Line 86, correct to “thereby”
  5. Line 100, correct to “Figure”
  6. Line 103, correct to “a dimer of”
  7. Line 106, B. cereus should be in italics
  8. Line 137m “8” should be subscript
  9. Line 164, “..subunits with respect..”
  10. Lnie 180, correct to 0.7 (not 0,7)
  11. Lne 206, correct to “cysteines”
  12. Lilne 257, correct “than” to “but"
  13. Line 276, remove one of the “only”
  14. Line 296, correct to “bonds”
  15. Lines 304-305, perhaps “..rBAT-ED, and suggest a…”
  16. Legend to Figure 3, please note that most places in the text HAG (not HaG) is used and conform the name throughout the manuscript
  17. Line 339, correct to “commercial”
  18. Line 351, correct to “knowledge”
  19. Line 358, add subscript “8” to the (β/α) barrel
  20. Line 361, correct to “..similarly to in α-amylases”
  21. Line 362, correct to “superdimers”
  22. Line 365, correct to “proven”
  23. Please note that the numbering of the reference list, should be corrected. Also all Latin names should be in italics
  24. Ref #39, correct to MacGregor
  25. Ref #60, correct to “oryzae”

Author Response

Dear reviewer,

thanks a lot for your deep review. It has been very helpful. We accepted almost all of the advices and changes and they are highlighted in yellow in the text. Below I attached some comments.

We accepted almost all of the advices and changes and they are highlighted in yellow in the text. Below I attached some comments.

Thank you,

Joana Fort

  1. We reviewed all the superscript of b0,+AT in the text that should be on superscript. This is the name for the transport system of this LAT, 0 for neutral and + means basic amino acids. We define b0,+AT as the LAT transporter associated to rBAT in the introduction (line 34). We avoid a deep explanation about transport systems, because there are many and it is not really the objective of the review. Nevertheless, I added a short explicative sentence in line 83.
  2. Following the advice we substitute “superfamily” by “family” and “subgrup” by “subfamily”.
  3. We rephrase line 109 as suggested with “..catalyze the action on α-glucosidic..”
  4. We rewrite the sentence with "the vicinities" line 112-114
  5. I rephrased lines 129-133 to do it more understandable about how many proteins and subunits form each transporter.
  6. Introduced full name for ASKA

All the minor points have been accepted and changed in the text (highlighted in yellow), but we want to comment some:

    1. Line 257, correct “than” to “but" -I didn't change it because it change the meaning for me. In fact is not the only one that is present, is the only one that is needed.
    2. Lines 304-305, perhaps “..rBAT-ED, and suggest a…” We changed to "rBAT-ED that can even lead to a total lack of α-glucosidase..."
    3. Legend to Figure 3, please note that most places in the text HAG (not HaG) is used and conform the name throughout the manuscript. I changed all the text to HaG, which is the name accepted in the literature.
    4. I changed all super-dimer in the text to superdimer.
